# Graphene/Polymer Nanocomposites: Preparation, Mechanical Properties, and Application

**DOI:** 10.3390/polym14214733

**Published:** 2022-11-04

**Authors:** Se Jung Lee, Seo Jeong Yoon, In-Yup Jeon

**Affiliations:** Nanoscale Sciences and Technology Institute, Department of Chemical Engineering, Wonkwang University, 460 Iksandae-ro, Iksan 54538, Jeonbuk, Korea

**Keywords:** graphene, reinforcement, nanocomposite, mechanical properties, application

## Abstract

Although polymers are very important and vastly used materials, their physical properties are limited. Therefore, they are reinforced with fillers to relieve diverse restrictions and expand their application areas. The exceptional properties of graphene make it an interesting material with huge potential for application in various industries and devices. The interfacial interaction between graphene and the polymer matrix improved the uniform graphene dispersion in the polymer matrix, enhancing the general nanocomposite performance. Therefore, graphene functionalization is essential to enhance the interfacial interaction, maintain excellent properties, and obstruct graphene agglomeration. Many studies have reported that graphene/polymer nanocomposites have exceptional properties that enable diverse applications. The use of graphene/polymer nanocomposites is expected to increase sustainably and to transform from a basic to an advanced material to offer optimum solutions to industry and consumers.

## 1. Introduction

Polymers are important materials in modern society as they are relatively inexpensive and easy to process compared to other materials (e.g., metals). However, their inadequate physical properties restrict their application in diverse areas. Therefore, polymers are usually enhanced with fillers of diverse sizes to relieve various restrictions and expand their application areas [1,2,3]. Nanoscale fillers, which refer to the size of the dispersed phase being less than 100 nm, have at least one characteristic length scale in the order of nanometers and vary essentially from isotropic to highly anisotropic sheet- or needle-like morphologies [3,4,5]. These were divided into three major types according to the nanoscale filler dimensions. First, two dimensions (2D) are on the nanometer scale, while the second is larger, forming an elongated one-dimensional (1D) structure that includes nanofibers or nanotubes (e.g., carbon nanofibers and nanotubes [6] or halloysite nanotubes [7]). Nanoscale fillers are iso-dimensional low-aspect-ratio nanoparticles (e.g., spherical silica [8], semiconductor nanoclusters [9], and quantum dots [10]). Third, two-dimensional (2D) nanoscale fillers (e.g., layered silicate [11], graphene [5,12], or MXene [13,14]) have the form of sheets that are one to a few nanometers and hundreds to thousands of nanometers thick.

Graphene has proven to be the most powerful material in the world, and has increasingly attracted attention as a promising candidate to substitute the position of carbon-based materials in the enhancement of the mechanical, electrical and thermal properties of polymers. Graphene is a one-atom-thick carbon layer arranged in a honeycomb structure. Additionally, several reports have introduced graphene into polymers for graphene/polymer nanocomposites [15,16,17], as graphene enhances the performance of polymers, including the mechanical strength, electrical conductivities, thermal stabilities, electro-chemical activity, and impart gas barrier properties [18,19,20,21,22,23]. However, the use of graphene has proven challenging due to the complicated preparation process, low solubility, and agglomeration by van der Waals interactions [24]. To resolve this, graphene-like materials with similar structures have been prepared from graphite or other carbon sources to maintain the several advantages of graphene while possessing oxygen-containing functional groups. Therefore, a method of developing new polymer/graphene nanocomposites and improving their physical properties is attracting attention.

## 2. Graphene

### 2.1. Preparation

Several studies have reported a variety of graphene synthesis techniques for pragmatic applications (Table 1) [25]. Each contributes to the diverse properties of the final material and has many possibilities for mass production. Thus, diverse graphene preparation processes were evaluated and compared with graphene quality (or purity) and the process (scalability, cost, and yield). Each method has its own characteristics; therefore, the choice of production method should be executed each time, according to the graphene applications.

#### 2.1.1. Mechanical Exfoliation

Mechanical exfoliation is the simplest process for preparing standalone graphene [26]. In this technique, graphite is repeatedly exfoliated utilizing tape and then transferred to a substrate. As a result, this can yield the highest quality, but is only used for lab scale and prototyping, as it is impossible to scale-up the process.

#### 2.1.2. Chemical Vapor Deposition (CVD)

CVD is the most beneficial process for fabricating high-quality single-layer graphene for use in various devices [27]. Graphene with a large area can be fabricated by exposing diverse hydrocarbon precursors to a metal at high temperatures. The unique mechanism for graphene formation based on the metal substrate begins with the growth of carbon atoms nucleated on the metal after hydrocarbon decomposition, and then the carbon nuclei develop gradually into large domains. Graphene transfer from a metal to other substrates is very difficult due to its chemical inertness, which causes defects and wrinkles in the final graphene. Additionally, thermal fluctuations affect graphene stability. Therefore, despite its complicated process and high energy demands, CVD is the most beneficial process for large-area graphene.

#### 2.1.3. Chemical Oxidation

Graphite oxide (graphene oxide or GO) was created via the Staudenmaier and Hummers methods. For example, graphite was oxidized with strong oxidative reagents (e.g., KClO_3_, KMnO_4_, and NaNO_3_) and concentrated H_2_SO_4_, HNO_3_, or their mixture [33]. Among these, the Hummers method is the most popular, wherein KMnO_4_ and NaNO_3_ are used as oxidizing agents for the oxidized graphite in the presence of concentrated H_2_SO_4_ (Figure 1). Generally, the safer and more scalable Hummers method is used to generate GO. GO contains several oxygen-containing functional groups (e.g., −OH, −COOH, −O−, and −C=O). Graphite is transformed into GO, which accompanies the exfoliation of the graphite layer. GO is readily dispersive in water and organic solvents and has poor electrical insulation and thermal instability properties. Therefore, to recover electrical and thermal properties, the reduction reaction for GO, termed as reduced graphene oxide (rGO), is required. Unfortunately, the complete reduction of GO to graphene has not occurred yet.

#### 2.1.4. Liquid-Phase Exfoliation

To obtain graphene, liquid-phase exfoliation necessitates three methods: (1) dispersion in a solvent or surfactant, (2) graphite exfoliation, and (3) purification through the separation of graphene and solvent removal [28,29]. The sonication time is a crucial factor as longer sonication time results in a lower graphene layer. Post sonication, the graphite and thicker graphene were removed by ultracentrifugation. Thus, the characteristics of graphene (e.g., yield and number of layers) can be controlled by varying the amount of graphite, sonication time, and centrifugation speed. Owing to poor contact between the graphene sheets, the electrical properties of the produced graphene were similar to those of GO. Additionally, due to very low graphene solubility, the use of a large amount of solvent increases the cost and is ecologically unfriendly.

#### 2.1.5. Electrochemical Exfoliation

Electrochemical exfoliation is generated by the anodic oxidation of the graphite-based electrode. Graphene with a few layers to the anode electrode is created, but its yield is very low and its properties are similar to those of GO, which comprises several oxygen-containing functional groups [30]. This technique offers a single step with an easy operation process that requires a production time over a period of minutes per hour compared with other methods that take a very long time (a few days) for preparation and purification. Additionally, the use of liquid electrolytes or aqueous surfactants makes the process eco-friendly, and that of LiClO_4_ as an electrolyte can prepare GO-like materials, preventing dangerous and toxic chemical materials from producing GO.

#### 2.1.6. Mechanochemical Reaction

To prepare large-quantity, high-quality, edge-selectively functionalized graphitic nanoplatelets (EFGnPs), a mechanochemical reaction was activated by ball milling (Figure 2) [31]. High-speed spinning metal balls (kinetic energy) are utilized to break the graphitic C−C bond that generates the chemical reaction of broken edges (C−X bond formation, X = heteroatom functional groups) and the physical delamination of graphitic layers into GnPs (graphene nanoplatelets, graphitic nanoplatelets, or GNP(s)). The X groups acted as physical wedges, preventing the GnP restacking. The graphitic C−C bond break induced by the kinetic energy of high-speed metal balls generates active carbon species, which have sufficient reactivity to combine with the appropriate reactants, which can be in diverse phases including vapors (CO_2_ [31], N_2_ [35], F_2_ [36], Cl_2_ [35], NF_3_ [37], and propylene [38]), liquids (SO_3_ [39], Br_2_ [35], heptane [40], and styrene [41]), solids (I_2_ [35], red P [42], S_8_ [43], Te [44], and Sb [45]), or their mixtures (CO_2_/SO_3_ [39] and SO_3_/I_2_ [46]). EFGnPs are oxygen-containing and comprise the desired functional groups or atoms at the edges. The oxygen-containing functional groups at the edges led to good EFGnP solubility in diverse solvents, allowing improved processability. 

### 2.2. Analysis Techniques

The unique properties of graphene have been verified by diverse analytical techniques. The atomic structure of graphene was verified by transmission electron microscopy (TEM) (Figure 3a), which is helpful for analyzing diverse structures [28]. Scanning tunneling microscopy (STM) offers information regarding graphene morphology and electronic properties in three dimensions (Figure 3b) [47]. Atomic force microscopy (AFM) was utilized to verify the number of layers as the graphene layer is 0.4 nm thick [48]. The specific surface area of graphene was estimated using the Brunauer–Emmett–Teller (BET) method with N_2_ adsorption–desorption isotherms [49]. X-ray diffraction (XRD) verified the exfoliation and intercalation into graphene layers and graphene formation (Figure 3c) [50]. X-ray photoelectron spectroscopy (XPS) was used to verify the chemical structure (e.g., chemical modification or functionalization) (Figure 3d) [50]. Raman spectroscopy was utilized to verify the number of layers and functionalization of graphene or graphene-like materials [25] which can recognize monolayer graphene, wherein the intensity of the 2D band is at least twice that of the G band (Figure 3e). Additionally, the 2D band of the two-layer graphene moved to a higher wavenumber, and the intensity of the 2D band was lower than that of the G band. Increasing graphene layers broadened the 2D band and developed a shoulder. UV–vis spectroscopy verified graphene dispersion in diverse solvents using Beer’s law and the linear relationship between absorbance and solution concentration (Figure 3f) [50].

### 2.3. Properties

The tremendous attention on graphene has resulted from its exceptional properties, which makes it an attractive material with enormous potential for applications in diverse industries and devices [32]. Its unique physical, mechanical, and electrical properties (e.g., specific surface area (2630 m^2^/g), intrinsic mobility (200,000 cm^2^/V·s), tensile strength (130 GPa), Young’s modulus (~1.0 TPa), thermal conductivity (~5000 W/m·K), optical transmittance (~97.7%), and electrical conductivity (~10^6^ S/cm)) have drawn lots of attention [48,53,54,55]. However, these characteristics depend on the graphene purity and thickness, which are major factors that determine the properties of graphene-based materials.

### 2.4. Graphene Functionalization

The intrinsic graphene properties are reduced dramatically as GO includes oxygen-containing functional groups and diverse defects on the basal plane and at the edges, and aggregation, which is derived from the high specific surface area and the strong van der Waals force between graphene sheets [56,57,58]. Proper GO functionalization inhibits aggregation, maintains excellent properties, and assigns new characteristics. Generally, GO functionalization can be divided into covalent and noncovalent methods (Figure 4) [59]. 

#### 2.4.1. Covalent Functionalization

Covalent functionalization introduces new functional groups to enhance the performance of graphene and graphene-like nanomaterials. GO comprises a variety of oxygen-containing functional groups (e.g., −COOH, −OH, −C=O, and −O−) that can be applied to general chemical reactions (e.g., isocyanation, carboxylic acylation, epoxy ring opening, diazotization, and addition) [60,61]. Additionally, other functionalizations, including carbon skeleton, diazotization, Diels–Alder reaction, and click chemistry reaction, can be conducted using the C=C bond [62,63,64]. Therefore, a double-bond addition reaction with C=C is performed to form a new C−C single bond, which is linked to the benzene derivative with a reactive functional group by a sigma bond. However, covalent graphene functionalization changes the hybridization from *sp^2^* to *sp^3^*, resulting in *π*-conjugation bond damage.

#### 2.4.2. Non-Covalent Functionalization

Non-covalent functionalization can be generated through supramolecular chemistry including *π–π* interactions, hydrophobic forces, hydrogen bonding, van der Waals forces, ionic bonding, and electrostatic effects [65,66]. The greatest advantage of non-covalent functionalization is the maintenance of the bulk structure and specific properties of graphene (or GO) and to increase GO dispersibility and stability. However, its weakness is that other substances (e.g., solvents and surfactants) are included.

## 3. Preparation of Graphene/Polymer Nanocomposites

### 3.1. Solution Method

The solution method is the most vastly used technique to prepare polymer nanocomposites on the laboratory scale due to its high utility through the use of diverse solvents and graphene along with a fast and simple process [67,68,69]. The polymer was dissolved in an appropriate solvent. The two substances were then blended using simple mixing, shear mixing, or ultrasonication for graphene dispersal in the polymer matrix. It is crucial to ensure solvent removal for nanocomposite manufacturing and homogeneous dispersion during the process. This is because the premier properties are significantly affected by the remaining solvent as they can plasticize the nanocomposites and exist at the surface. In the solvent evaporation process, it is important to increase the solubility in the solvent due to the re-aggregation of graphene [32]. Additionally, very low solubility of some polymers in common solvents and the use of large solvent quantities of solvents are some of the limitations of this method. 

### 3.2. Melting Method 

The melting method is an industrial-friendly process to predominantly produce thermoplastic-based nanocomposites owing to its rapid and economic process [2,70]. This significantly affects polymer disentanglement in the molten state. Therefore, the polymer chains move freely and mix thoroughly during the molten state. The polymer nanocomposites prepared by melt mixing exhibited satisfactory dispersion of the fillers. However, the manufacturing process for the mixing temperature should be careful because the polymer may degrade at a high temperature [25]. Additionally, high shear forces are required for efficient mixing that leads to the formation of folds/wrinkles or even breaking of the nanoplatelets, decreasing the effective modulus. After melt mixing, additional steps including hot pressing and injection molding were necessary, which are affected by the dispersion, structure, and orientation of the fillers.

### 3.3. In-Situ Polymerization

In situ polymerization enables the filler to be grafted onto the polymer, which improves the compatibility and interface between the constituents [71,72,73,74]. The monomer and graphene were dispersed in an appropriate solvent and ultrasonicated for homogeneous dispersion. An initiator was then incorporated to the mixture for polymerization. However, the polymerization process itself generally increases the viscosity of the mixture, making further processing difficult, which eventually limits the loading fraction. In some cases, this method requires the use of a solvent and requires additional refinement steps to remove the solvent [32]. Its advantages include ease of manipulation, simplicity, scalability, low cost, and low environmental anxiety.

### 3.4. Electrochemical Reaction

Electrospinning is used for preparing fine nanofibers with average diameters ranging from nanometers to micrometers [2,75]. Electrospun fibers have attractive characteristics, including a high surface/volume ratio, which induces a low density, high pore volume, and exceptional mechanical strength. Additionally, process factors including the type, molecular weight, viscosity, solvent, applied voltage, needle-to-collector distance, and flow rate play a critical role in obtaining the desired properties [76,77].

Electrodeposition is a simple and fast method for producing nanocomposites via electrochemical reactions [2]. It involves the electro-polymerization of a graphene/polymer nanocomposite from the monomer, doping matter (if needed), and GO. Electrodeposition occurs at a specific potential and stops when a pertinent amount of charge has passed. Therefore, the nanocomposite encompassed the electrode surface.

## 4. Mechanical Properties

Carbon-based polymer nanocomposites have been used to improve the mechanical performance of pure polymers by increasing the interaction between polymers and carbon materials. Among them, graphene/polymer nanocomposites exhibit significantly improved mechanical properties compared to those of pure polymers [32,78,79]. The merits of graphene compared to other fillers enable smooth changes in the properties of nanocomposites at very low percolation thresholds due to the very high aspect ratio of graphene. GO, which comprises oxygen-containing functional groups, is hydrophilic and can interact well with polymers. Therefore, it can improve the mechanical parameters (e.g., Young’s modulus, fracture toughness, fracture time, thermal stability, electrical conductivity, and gas barrier properties) [2,80,81]. The incorporation of graphene within the matrix increases the crosslinking of the polymer chains, which leads to an increase in the mechanical properties owing to the exceptional mechanical properties of graphene (tensile strength of ~130 GPa and Young’s modulus of ~1 TPa) [82,83]. However, due to the filler, the movement of the polymer chain is inhibited, so the tension toughness and strain of nanocomposites may tend to decrease. Therefore, many papers have reported that the mechanical properties of graphene/polymer nanocomposites are considerably enhanced compared to those of pure polymers (Table 2). The table was assembled according to the type of graphene and polymer.

### 4.1. Graphene/Polyethylene Nanocomposites

In the case of ultrahigh-molecular-weight polyethylene (UHMWPE)/thermally reduced graphene oxide (TrGO), the tensile strength and Young’s modulus improved with an increase in the TrGO content due to the exceptional inherent TrGO strength and stiffness [84]. In the case of 1 wt.% TrGO, the tensile strength of the pure UHMWPE increased from 1.1 to 3.1 GPa (182% increase) and the Young’s modulus improved by over three times. The drawing ratio is also a factor that improves the mechanical properties of the composite. When the composite film is not drawn, the filler is randomly distributed within the PE matrix. As the drawing ratio increases, the filler cluster expands so that the TrGO flakes are aligned along the drawing direction, and the size of the cluster decreases, which means that the flakes are exfoliated. The exfoliated flakes increase the specific area in contact with the PE matrix molecule and the TrGO fillers and improve load transfer from PE to TrGO. Additionally, the significant increase in the mechanical properties at low TrGO content was ascribed predominantly to the uniform TrGO dispersion and very high specific surface areas that help transfer the load from the polymer matrix to the strong fillers. 

High-density polyethylene (HDPE)/graphene nanocomposites were prepared directly from HDPE and graphene using a twin-screw extruder. HDPE stiffness increased significantly with increasing graphene content, with a maximum Young’s modulus (187% increase) [86]. Therefore, the tensile strength increased to 34%, whereas the elongation at break reduced to 91%. The improvement in tensile strength without decreasing the elongation is due to the exceptional mechanical strength of graphene, which is a uniform dispersion in the matrix and strong interfacial interactions [140]. 

Heptene-functionalized graphitic nanoplatelets (HGN) produced by ball milling were incorporated as reinforcing fillers for LLDPE (linear low-density polyethylene) [40]. The tensile strength and Young’s modulus of the HGN/LLDPE nanocomposites were higher than those of pure LLDPE (Figure 5a,b). The HGN/LLDPE_5 (5 wt.% HGN content) displayed the best mechanical properties among the pure LLDPE and HGN/LLDPE nanocomposites. Therefore, the tensile strength, yield strength, Young’s modulus, and tensile toughness of the HGN/LLDPE_5 nanocomposites were enhanced to 43.8, 39.4, 39.5, and 125.8%, respectively, compared with those of pure LLDPE. Comparing the fracture surfaces, pure LLDPE appeared smooth and plain with wave-shaped morphology, but HGN/LLDPE _5 displayed rough and canyon-like structures (Figure 5c,d), indicating that the specimen was ripped steadily by an accumulated stress that transferred efficiently from HGNs to LLDPE due to good dispersion of the HGNs into LLDPE.

### 4.2. Graphene/Polypropylene Nanocomposites

Polypropylene (PP)/reduced graphene oxide (PP/rGO_X) nanocomposites with low rGO concentrations (X = rGO content, 0.05–1.0 wt.%) were created via twin-screw extrusion [141]. The tensile strength and Young’s modulus of PP/rGO_1 increased by 11.15 and 41.45%, respectively, compared with those of pure PP. The enhanced mechanical properties were induced by the exceptional rGO dispersion due to the combination of shear and extensional flows during extrusion and good compatibility between rGO and PP. 

PP/graphene (polypropylene coated graphene) nanocomposites with exfoliated PP-coated graphene, which can prevent the restacking of graphene sheets during melt blending, demonstrated exceptional interaction and effective load transfer between the polymer and graphene [93]. In the case of 0.1 wt.% graphene, the yield strength, tensile strength and Young’s modulus were enhanced to 30 MPa (by 36%), 33 MPa (by 38%), and 1.25 GPa (by 23%), respectively, compared with pure PP. Moreover, the elongation at break remained almost unchanged, indicating that the nanocomposite toughness was not reduced. In the case of 1.0 wt.% graphene content, the yield strength (38 MPa), tensile strength (37 MPa), and Young’s modulus (1.76 GPa) increased by 75%, 54%, and 74%, respectively, compared with pure PP. This is ascribed to the uniform graphene dispersion and effective load transfer from the polymer to graphene through interfacial interactions.

Edge-propylene graphitic nanoplatelets (EPGnP/PP)/PP_X nanocomposites were prepared utilizing a solution method [38]. EPGnP/PP_5 (EPGnP content of 5 wt.%) displayed that tensile strength, yield strength, and Young’s modulus increased to 90.5, 90.8, and 249.5%, respectively, compared to pure PP (Figure 6a,b). This is due to the molecular-level dispersion of the EPGnPs into the PP matrix, and the strong interfacial interaction between EPGnP and PP through the EPGnP propylene functional groups. It verified the uniform dispersion of EPGnPs into the PP matrix using HR-TEM (Figure 6c,d). However, as the content of EPGnP increased, the tensile toughness of the EPGnP/PP_X nanocomposite decreased because the strain of the EPGnP/PP_X nanocomposite decreased gradually because the addition of multi-dimensional fillers limits the movement of the matrix. Although the filler is uniformly dispersed within the matrix, there is little elongation, and EPGnP behaves like a physical cross-linking point within the PP matrix. 

### 4.3. Graphene/Epoxy Nanocomposites

The epoxy/GNP_X nanocomposites manufactured by melt mixing demonstrated an increase in the GNP content, tensile strength, and Young’s modulus of the nanocomposite [94]. In epoxy/GNP_10 (GNP content:10 vol.%), the tensile strength and Young’s modulus increased to 1.0 ± 0.08 MPa (66.7) and 26 ± 1.7 MPa (1344%), respectively, compared with pure epoxy. The tensile strength was fixed at 5 vol.% GNP. The tensile strength increased to 1.4 ± 0.07 MPa at 5 vol.% GNP and then was decreased to 1.0 ± 0.08 MPa at 10 vol.%, indicating an increase of 133.3% at 5 vol.% and a decrease of 28.6% at 10 vol.%. The increased tensile strength at low graphene content is attributed to the exceptional graphene dispersion and powerful mechanical interaction between graphene and the polymer. 

Diverse epoxy nanocomposites with pristine- and Triton-graphene (graphene functionalized non-covalently with Triton-X100) have been prepared [100]. The corresponding stress of the nanocomposite was higher than that of pure epoxy for the same strain, indicating that the nanocomposite modulus was improved by the incorporation of pristine- and Triton-graphene. At the breaking point on the S–S curves, the strength and elongation at break of the nanocomposite with Triton-graphene were obviously higher than those of pure epoxy and pristine graphene-filled epoxy nanocomposite. The tensile strength of pristine graphene was almost unchanged compared to that of pure epoxy, whereas that of the Triton-graphene (0.1 wt.% loading) was greatly enhanced by 57% (from 52.98 ± 5.82 to 83.43 ± 5.90 MPa), although the elastic modulus still demonstrated a similar increase to that of the pristine graphene nanocomposites. As depicted in the SEM images of the Triton-graphene/epoxy nanocomposite fracture surface, well-dispersed graphene and a relatively good graphene/polymer interface effectively improved the load transfer efficiency with the polymer and thus increased the tensile strength. 

### 4.4. Graphene/Poly(Vinyl Alcohol) Nanocomposites 

PVA/GO-X (X = GO content) nanocomposites were prepared utilizing the solution method [110]. Compared with pure PVA, a maximum increase of approximately 50% in the tensile strength was observed for the nanocomposite (X = 0.3 and 2.0). Additionally, its elongation at break or failure strain increased to 13–22%, indicating that the exceptional GO dispersion at a lower content led to a significant increase in the mechanical properties. However, the elastic modulus of the PVA/GO-2 was larger than PVA/GO-0.3 nanocomposite due to the higher GO content, which possesses a high elastic modulus.

PVA/GO-X (X = GO content) nanocomposites were prepared via in situ polymerization [112]. The tensile strength and Young’s modulus of the PVA/GO-X nanocomposites improved with increasing GO content at low content. Therefore, the tensile strength and Young’s modulus of PVA/GO-0.04 were enhanced from 50.8 MPa and 2123 MPa compared with pure PVA (42.3 MPa and 1477 MPa). However, elongation behavior decreased slightly from 215% to 211.5% due to the strong H-bonding interaction between the GO surface and the PVA matrix, that prevents and reduces the movement of polymer chains. The enhanced mechanical properties at low GO contents are related to the combination of GO with excellent tensile strength and Young’s modulus, homogeneous GO dispersion, and strong hydrogen bonding between the oxygen-containing functional groups of GO and PVA chains.

### 4.5. Graphene/Polyurethane Nanocomposites

PU-GX (X = graphene content) nanocomposites were prepared with waterborne biodegradable PU and graphene at diverse graphene contents (0, 1, 3, and 5 wt.%) [118]. Graphene enhanced the mechanical properties of PU-GX nanocomposites. The tensile strengths of PU, PU-G1, PU-G3, and PU-G5 nanocomposites were approximately 35.3, 23.3, 22.9, and 11.4 MPa, respectively. Additionally, the elongation of PU-G5 (317%) decreased compared to that of pure PU (640%) as the incorporation of graphene to pure PU induces the nanocomposites to be less elastic.

PU/graphene nanofibers were obtained by electrospinning a PU/graphene solution comprising GO, f-GO (PCL-functionalized GO), and r-GO at diverse concentrations (0, 0.1, 0.5, and 1.0 wt.%) [117]. The modulus and breaking stress of the PU/graphene nanocomposite nanofibers were higher than those of pure PU. The breaking stress of the nanocomposite nanofiber at 0.1 wt.% f-GO was 9.3 MPa, which is 1.2 times higher than pure PU nanofibers (7.8 MPa). The nanofiber modulus increased with increasing graphene content. The Young’s moduli of the nanofibers with 1 wt.% f-GO displayed 41.4 MPa, whereas that of pure PU nanofiber webs was 31.5 MPa. Thus, f-GO was very well dispersed in the polymer matrix compared with GO and r-GO and functioned for excellent load transfer. 

Hyperbranched aromatic polyamide-functionalized graphene sheets (GS-HBA) were prepared with 3,5-diaminobenzoic acid (DABA) as a monomer [122]. GS-HBA demonstrated good dispersion in thermoplastic polyurethane (TPU) and strong adhesion with TPU via hydrogen bonding, which efficiently enhanced the load transfer from TPU to the graphene sheets. 

As the GS-HBA content increased, the stress gradually increased, and the strain steadily decreased. The S–S curve of the 15 wt.% GS-HBA nanocomposite was similar to pure TPU. The highest Young’s modulus of the nanocomposite with 15 wt.% GS-HBA was 1.2 GPa, that is one order of magnitude higher than pure TPU (0.09 GPa). The yield tensile strengths of nanocomposites with 15 wt.% GS-HBA (25 MPa) increased about seven-fold compared with pure TPU (3.4 MPa), that is comparable to UHMWPE (about 22 MPa). Initially, the ultimate tensile strength increased from 18 to 37 MPa, and then decreased slightly to approximately 30 MPa when the GS-HBA content increased to over 2.5 wt.%. GS-HBA may advance strain hardening and strain-induced crystallization of soft segments (SS) owing to hydrogen bonding. As the GS-HBA content increased, the strain at the break of the nanocomposites decreased. 

### 4.6. Graphene/Polystyrene Nanocomposites

PS/FLG-X (X = FLG content) nanocomposites were prepared by combining graphene produced via liquid-phase exfoliation with polystyrene [127]. The Young’s moduli of the nanocomposites with FLG from 0.1 to 0.9 wt.% increased from 55.4 to 59.8% compared to pure PS (0.202 GPa) and achieved this at the peak value of the PS/FLG-0.9 composite. This is because FLG impedes the chain mobility of the host polymer. An increase in the ultimate tensile strength was observed based on the graphene value. The smallest value is at PS/FLG-0.1 and the highest is at 13.98 MPa of the PS/FLG-0.9 nanocomposite; the overall improvement in UTS was 74.75%. The polymer chain interacted with graphene and formed a bond, increasing the tensile strength of the nanocomposite.

Edge-styrene graphitic nanoplatelets (StGnPs) were prepared directly using a mechanochemical reaction with graphite and styrene, and StGnP/PS-X (X = StGnP content) nanocomposites were prepared utilizing solution methods [41]. The mechanical properties of the StGnP/PS nanocomposite depended on the StGnP content. The StGnP/PS-5 nanocomposite exhibited the best tensile strength (11.54 MPa), Young’s modulus (809.4 MPa), yield strength (3.67 MPa), and tensile toughness (0.89 MPa) (Figure 7). The considerably enhanced mechanical properties of the StGnP/PS nanocomposites compared with those of pure PS can be ascribed to the stress being transferred comfortably from PS to StGnPs through the molecular-level StGnP dispersion into PS and strong interfacial interaction between the StGnPs and PS. 

### 4.7. Graphene/Poly(Vinyl Chloride) Nanocomposites

Multi-layer graphene (MLG) PVC nanocomposites (MLG/PVC-X, X = MLG content) were prepared by in situ polymerization and melt mixing [131]. The yield strength and Young’s modulus of the MLG/PVC nanocomposites strongly depended on the MLG content. The yield strength and Young’s modulus first increased and then decreased with MLG incorporation. The yield strength and Young’s modulus of the MLG/PVC-0.3 nanocomposites increased by 11 and 8%, respectively, compared to those of pure PVC. The exceptional dispersion of MLG and strong interactions led to the efficient transfer of the load and interruption of the motion of PVC chains, resulting in the excellent mechanical properties of the nanocomposites.

PVC nanocomposites embedded with graphene nanoplatelets (GNP/PVC) were prepared utilizing a solution method [132]. The GNP/PVC nanocomposites exhibited a far higher elastic modulus and strength than pure PVC. For the GNP/PVC nanocomposites (2.5 wt.% GNP), elastic modulus and ultimate tensile strength were 1.42 GPa (108.77%) and 24.0 MPa (100%), respectively, compared to pure PVC (5.47 GPa and 12.0 MPa). The exceptional mechanical properties of the GNP/PVC nanocomposites were attributed to the strong interaction between GNP and PVC and high GNP modulus.

### 4.8. Graphene/Polyamide Nanocomposites

GO/nylon 6 was prepared via a solution-mixing method [136]. The incorporated graphene induced a noteworthy enhancement in the mechanical strength and Young’s modulus. At 1.0 wt.% graphene, the best yield strength (91 MPa) and Young’s modulus (1.90 GPa) increased to 18 and 28%, respectively, compared to pure Nylon 6.

Nylon 6/sulfonated graphene nanocomposites (NSG-X, X = SG content) were prepared via in situ ring-opening polymerization of ε-caprolactam and sulfonated graphene (SG) with polar sulfonic acid [137]. Compared with pure nylon 6, NSG-0.2 demonstrated an improvement in tensile, impact, and especially bending strength, due to the buildup effect of SG with high specific surface areas. The bending and impact strengths increased by 32.2 and 5.7%, respectively, compared to those of pure nylon 6. However, in the case of NSG-3.0 with the highest SG content, the mechanical properties decreased. The low mechanical properties are due to the low molecular weight of grafting nylon 6 chains on the SG sheets. It also decreased with the elongation at break of the NSG composites with increasing SG content, due to the high density of grafting and the inability molecular “slippage”. Therefore, mechanical properties may be maintained and improved by using an appropriate SG content.

### 4.9. Analysis Methods of Properties for Graphene/Polymer Nanocomposites

The mechanical properties and thermal stability of the graphene/polymer nanocomposite are influenced by the internal microstructure and crystallinity. Thermal properties of polymers can be confirmed with various measuring equipment (DSC, TGA, DMA etc.) and the T_m_, T_g_, T_c_ of nanocomposites can be found from the measurement data [142]. In particular, glass-transition temperature, T_g_, is one of the important factors in polymer treatment and production applications, and it depends on the molecular weight of the polymer chain and the spatial structure of the polymer [143].

Differential scanning calorimetry (DSC) is used to study the thermal behavior of the nanocomposites and to follow their melting and crystallization behavior [89]. Liang et al., [109] used DSC to compare T_g_ of pure PVA with the T_g_ of graphene/PVA nanocomposites. As a result, it can be observed that T_g_ of the graphene/PVA nanocomposite increased compared to pure PVA. The increase in T_g_ indicates that the polymer chain was limited by the H-bonding interaction. Load transfer is largely dependent on the interfacial interaction between the filler and the wrapped polymer matrix [144]. In addition, the molecular-level dispersion of graphene sheets in the PVA matrix and the large aspect ratio of graphene are also advantageous for stress transfer through graphene/PVA interfaces. Therefore, significant improvement in tensile stress and modulus of the nanocomposite was greatly improved. The crystallization temperature (T_c_) may also be obtained through DSC measurement. As the content of GNP in the PP/GNP nanocomposites increases, T_c_ shifted to a higher temperature, indicating that the presence of GNP as a nucleating agent in nanocomposites promotes the crystallization of PP [145,146]. The addition of GNP provides more surfaces in the nucleation process that facilitates the crystallization process [89].

The thermal stability and composition of the nanocomposites are investigated using thermogravimetric analysis (TGA). Additionally, TGA was used to confirm filler loading and evaluate the effect of filler loading on the thermal stability in nanocomposites. The degradation performance was improved as filler loading increased, due to the hindered effect of the filler upon the diffusion of oxygen and volatile products throughout the nanocomposite material. That is, it can be seen that the thermal stability of the polymer was improved by adding graphene [89].

Tensile test and dynamic mechanical analysis (DMA) have been conducted to evaluate the enhancement properties of nanofillers [147]. According to the storage modulus measured by DMA, it can be seen that the values of the IL-G/PU nanocomposites are higher than the value of the pure PU. The data of tan δ measured by DMA showed that as the ILG content increases, the T_g_ of the composites gradually transfers to higher temperatures [123]. The high specific surface area of the IL-G sheet provides an excellent contact area with the PU matrix, obtaining a strong interfacial attraction through π-π and/or cation-π as well as electrostatic interactions and van der Waals forces. Additionally, the presence of tertiary amine on the IL-G sheets may interact with the carbonyl group in the PU chain through hydrogen bonding [148,149]. The synergistic effects improve T_g_ of the PU nanocomposite by constraining the mobility of PU chains segment [123].

## 5. Applications

An optimized graphene content in a polymer matrix is typically used for structural reinforcement areas [150,151,152,153]. Owing to the potential multifunctional properties of polymers with graphene, graphene/polymer nanocomposites have been utilized in several areas. The noteworthy enhancement of the mechanical performance of polymers suggests the use of graphene in diverse applications to achieve a combination of high strength and light weight. Additionally, it has demonstrated a potential to enhance the safety, reliability, and cost-effectiveness of graphene/polymer nanocomposites, which can increase the range of material options in the aerospace, automotive, marine, sports, biomedical, and energy industries (Figure 8).

Owing to the outstanding structural strength and conductivity of graphene, polymer nanocomposites have become a potential candidate for aerospace applications [150]. Thermosetting polymers are predominantly used as matrices for aerospace as they are thermally stable, chemically inactive, and have reasonable mechanical and electrical properties. However, most thermosetting polymers are fragile at low temperatures due to their stiff cross-linked structures, which mean that they crack easily according to thermal-fatigue loading [154]. Therefore, to resolve various issues, it is essential to improve the properties of the matrix or its interfacial bonding. Diverse types of graphene can be utilized for this purpose. This will be extremely helpful in satisfying the normal requirements of aerospace applications.

Graphene/polymer nanocomposites have advantages in diverse environmental applications including water treatment, energy production, and contaminant sensing [150,151,155]. Graphene/polymer nanocomposites preserve them against corrosion, thus, they were useful as adsorbents for the removal of metal ions, organic materials, and gases in the aquatic environment, and were also utilized to eliminate water contaminants.

Research is being conducted on the use of graphene as a new and renewable energy source. In solar cells, graphene either acts as the active medium or as a transparent/distributed electrode material [151,156]. It is a highly flexible and stretchable electrode that can be used in diverse electrolytes.

Graphene possesses properties that are distinct from those used in biotechnological and bio-based applications. Owing to its large surface area, chemical stability, and functionalization feasibility, it is a promising candidate for tissue engineering, drug delivery, and DNA sequencing [157]. Furthermore, ultrasensitive measurement equipment to sensitively detect diverse biological molecules, including glucose, hemoglobin, cholesterol, and DNA, can be prepared using chemically functionalized graphene. Graphene membranes with nanopores are ideal for achieving higher permeate flux, higher selectivity, and increased stability by controlling the pore size and aspect. Graphene paints were concatenated with conductive ink, antistatic, magnetic shielding, and gas barriers. Thus, to expand the application areas of graphene/polymer nanocomposites, new graphene chemical modifications to control the properties of products have been reported.

## 6. Future Perspectives

Among the various reinforcement materials, graphene has excellent mechanical, thermal, and electrical properties, and is regarded as a better alternative to general nanofillers. Thus, graphene/polymer nanocomposites have properties that are extraordinary for diverse applications. Several studies on the exceptional graphene properties have revealed that it plays an imperative role in enhancing the specific properties of polymer nanocomposites, which can be utilized in diverse applications [32,150,158]. Other than in the above-mentioned application, in order to utilize graphene/polymer nanocomposites, improvements are required in graphene dispersion in a matrix, interfacial interaction, mass production and remarkable quality, etc. The interfacial interaction between graphene and the polymer and the homogeneous dispersion of graphene generally enhances nanocomposite properties. A suitable interaction between graphene and polymer guarantees the technological advancement of graphene/polymer nanocomposites, but is still insufficient. Therefore, the challenge is to maintain the inherent properties of graphene in the nanocomposite as much as possible, many people continue to strive to overcome this issue.

## 7. Conclusions

Current polymer nanocomposites are useful in various aspects. Many studies have been reported demonstrating that graphene/polymer nanocomposites can be applied in various fields through the chemical functionalization of graphene used as fillers. The uniform dispersion of graphene is one of the factors that can improve performance. Additionally, the interaction between the polymer and the filler, the surface area, filler loading, and dispersion are various factors that determine the quality of the nanocomposite. Despite numerous challenges related to graphene, a suitable interaction between graphene and polymers is yet to be developed. However, the manufacturing industry cannot neglect the use of graphene (or graphene-like materials) in commercial products. It is expected that it will be transformed from a typical to a high-performance material that can offer the best solutions for industry and consumers. Therefore, the business influence of graphene/polymer nanocomposites is growing continuously.

## Figures and Tables

**Figure 1 polymers-14-04733-f001:**
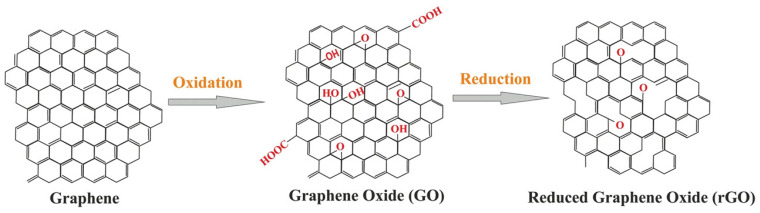
Conversion of graphene into GO and rGO. Reprinted from Ref. [34].

**Figure 2 polymers-14-04733-f002:**
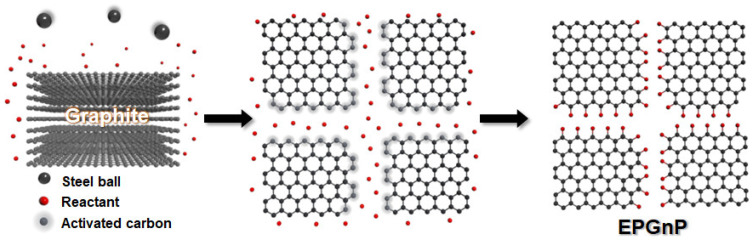
Conceptual diagram for preparation of EFGnPs by mechanochemical reaction.

**Figure 3 polymers-14-04733-f003:**
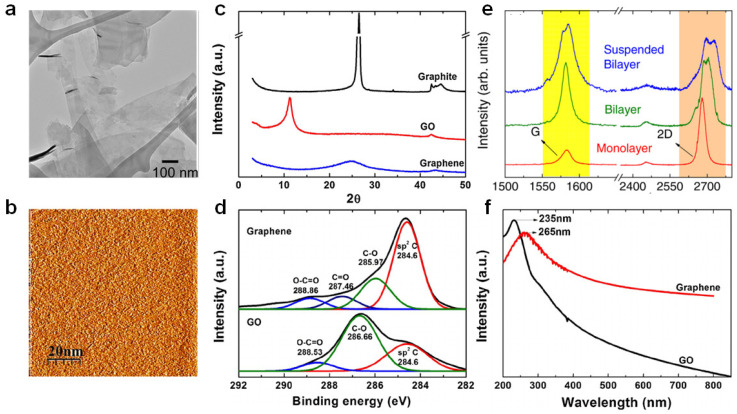
(**a**) Flakes of few-layer graphene on a holey carbon TEM grid. Adapted with permission from Ref. [51]. (**b**) STM images of graphene. Reprinted with permission from Ref. [47]. Copyright 2012 Hong et al. (**c**) XRD patterns of graphite, graphene oxide and graphene. Adapted with permission from Ref. [50]. Copyright 2014 Johra and Lee. (**d**) XPS spectra of GO and graphene [50]. (**e**) Raman spectra of monolayer, bilayer, and suspended bilayer graphene. Adapted with permission from Ref. [52]. Copyright 2011 AIP Publishing. (**f**) UV–vis spectra of GO and graphene. Adapted with permission from Ref. [50]. Copyright 2014 Elsevier.

**Figure 4 polymers-14-04733-f004:**
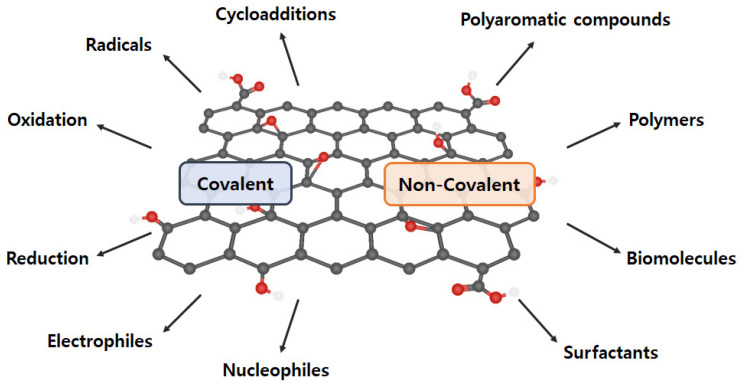
Various covalent and non-covalent graphene functionalization.

**Figure 5 polymers-14-04733-f005:**
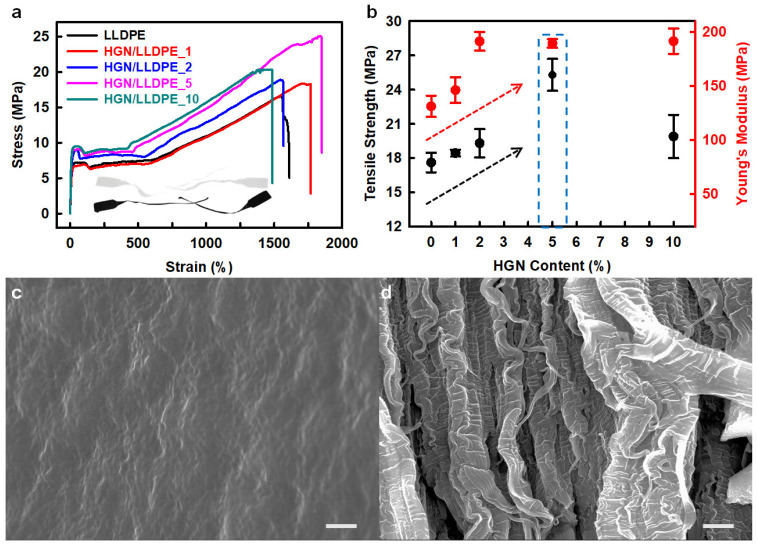
(**a**) stress–strain (S–S) curves of the pure LLDPE and HGN/LLDPE nanocomposites. (**b**) Tensile strengths and Young’s modules of the pure LLDPE and HGN/LLDPE nanocomposites according to HGN content. SEM images of the fractured surface: (**c**) pure LLDPE, and (**d**) HGN/LLDPE_5. Scale bars are 10 μm. Reprinted with permission from Ref. [40]. Copyright 2020 Elsevier.

**Figure 6 polymers-14-04733-f006:**
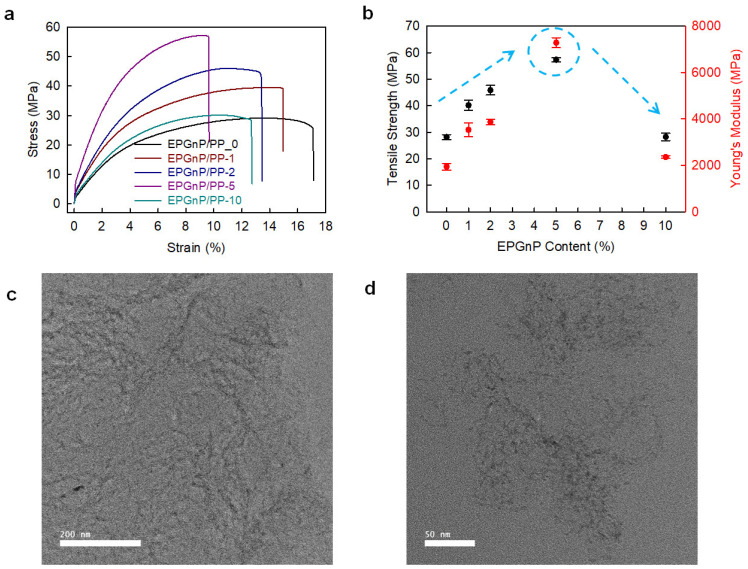
(**a**) S–S curves and (**b**) tensile strength and Young’s modules of EPGnP/PP_X nanocomposites. HR-TEM images of EPGnP/PP_5 nanocomposites: (**c**) low-magnification; (**d**) high-magnification. Reprinted with permission from Ref. [38]. Copyright 2021 Elsevier.

**Figure 7 polymers-14-04733-f007:**
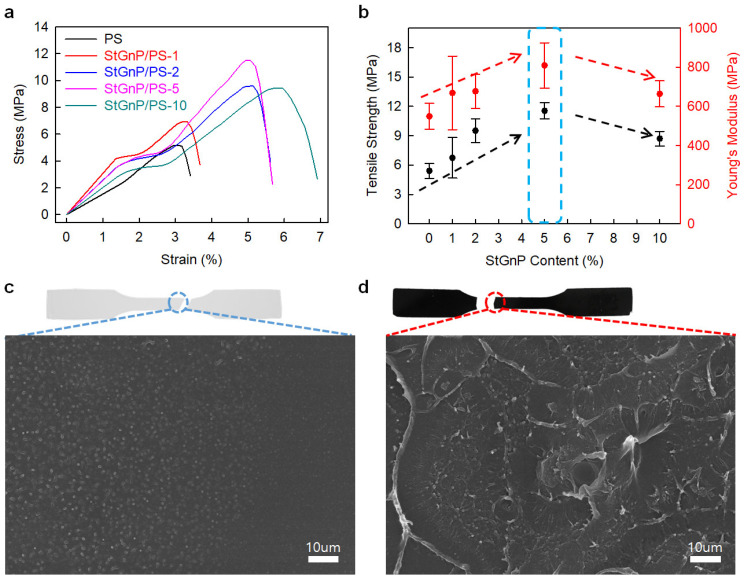
(**a**) S–S curves of pure PS and StGnP/PS nanocomposites. (**b**) Tensile strength and Young’s modulus of pure PS and StGnP/PS nanocomposites. SEM images of the fractured surfaces after the tensile test: (**c**) pure PS and (**d**) StGnP/PS-5 nanocomposite. Reprinted with permission from Ref. [41]. Copyright 2021 Kang and Kim.

**Figure 8 polymers-14-04733-f008:**
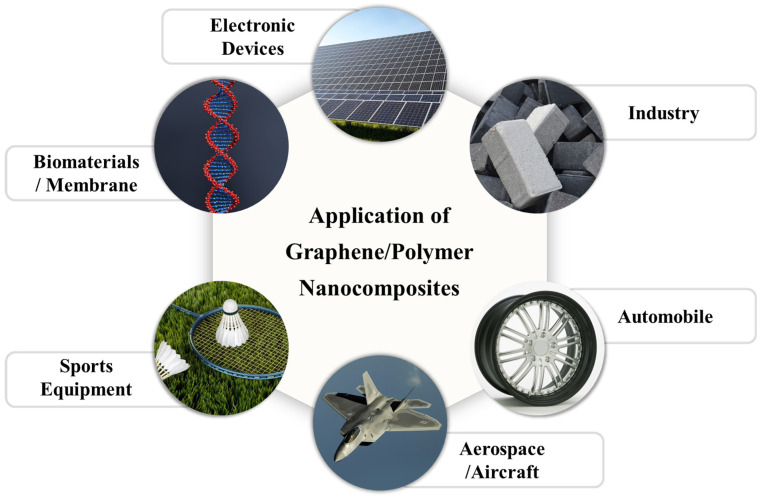
Applications of graphene/polymer nanocomposites.

**Table 1 polymers-14-04733-t001:** Advantages and disadvantages of the graphene preparation method.

Methods	Advantage	Disadvantage	Ref.
Mechanical Exfoliation	High-quality Simplest process	Small production scale	[26]
Chemical Vapor Deposition	High-quality Large-area graphene	Complicated process High energy demand	[27]
Chemical Oxidation (Hummers method)	Fast reaction Fewer defect	High contamination and degradation	[25]
Liquid-Phase Exfoliation	Mass production Upscaling production	Poor solubility Eco-friendly	[25] [28] [29]
Electrochemical Exfoliation	Single step Eco-friendly	Expensive	[25] [30]
Mechanochemical Reaction	Mass production High-quality Edge-selectively	High energy consumption	[31] [32]

**Table 2 polymers-14-04733-t002:** Mechanical properties of diverse graphene/polymer nanocomposites.

Graphene	Matrix	Process	Filler Loading	Tensile Strength (MPa)	Young’s Modulus (MPa)	Strain (%)	Ref.
TrGO	UHMWPE	Solution process	1 wt.%	3100	106,000		[84]
GNP	LLDPE	Solution process	5 wt.%	25.3	189.4		[40]
MGO	LLDPE	Solution process	3 wt.%	19.9			[85]
GNP	HDPE	Melt mixing	23 wt.%	34.84		7	[86]
TRG	HDPE	Polymerization	5.2 wt.%	12.9	624.4	9.7	[87]
GNP	PP	Melt mixing	10 wt.%		1963.2	18.20	[88]
GNP	PP	Melt mixing	5.5 wt.%		1900	7.00	[89]
GNP	PP	Melt mixing	5 wt.%	38		6.99	[90]
GNs	PP	Melt mixing	3 wt.%	61.57	2314.61	19.09	[91]
fGO	PP	Melt mixing	1 wt.%	38.7	562		[92]
Graphene	PP	Melt mixing	0.1 wt.%	33	1250	1150	[93]
Graphene	PP	Melt mixing	1 wt.%	37	1760	130	[93]
GNP	PP	Solution process	5 wt.%	55.85	7239	9.07	[38]
GNP	Epoxy	Solution process	10 vol%	1	26		[94]
Graphene	Epoxy	Melt mixing	0.5 wt.%	23.01	8000		[95]
GNP	Epoxy	Solution process	6 wt.%	53	3400	2	[96]
GNP	Epoxy	Melt mixing	6 wt.%	35.5		1.49	[97]
GNP	Epoxy	Melt mixing	4 wt.%	75.8		4.55	[98]
GNP	Epoxy	Solution process	0.3 wt.%	72.4	1990	8.2	[99]
fGr	Epoxy	Solution process	0.1 wt.%	83.43			[100]
rGO	PVA	Wet spinning	2 wt.%	867	15,900		[101]
GO	PVA	Solution process	0.3 wt.%	63			[102]
GO	PVA	Solution process	3 wt.%	110		36	[103]
rGO	PVA	Solution process	0.02 wt.%	45.6	162		[104]
rGO	PVA	Solution process	3.5 wt.%	29	520	22	[105]
GS	PVA	Solution process	1 wt.%	67.7		139	[106]
rGO	PVA	Solution process	0.7 wt.%	154	4900	5.1	[107]
GNs	PVA	Solution process	1.8 vol%	42	1040		[108]
GO	PVA	Solution process	0.7 wt.%	87.6	3450		[109]
GO	PVA	Solution process	2 wt.%	37.8	67.3	294	[110]
Graphene	PVA	Electrospinning	6 wt.%	19.2	638	113	[111]
GO	PVA	Polymerization	0.04 wt.%	50.8	2123	208	[112]
GNP	PVA	Electrospinning	1 wt.%	11		130	[113]
rGO	PVA	Electrospinning	2 wt.%	5.51	85.67		[114]
GNs	PU	Polymerization	2 wt.%	36.3		535	[72]
fGNP	PU	Polymerization	1.5 wt.%	23.4		6.7	[115]
fGO	PU	Solution process	0.4 wt.%	19.6		1035.3	[116]
fGO	PU	Electrospinning	1 wt.%	8.9	41.4	515.6	[117]
Graphene	PU	Solution process	3 wt.%	22.9	2.7	474	[118]
fGS	PU	Solution process	1 wt.%	11.9		448	[119]
fGNs	PU	Solution process	2 wt.%	20.2		138	[120]
TrGO	PU	Polymerization	2 wt.%	10.6	35.1	715	[121]
GSs	PU	Solution process	15 wt.%	25	1200	220	[122]
MrGO	PU	Solution process	0.608 wt.%	34.30		186.24	[123]
fGS	PU	Solution process	1 wt.%	40		590	[124]
GNP	PS	Solution process	5 wt.%	11.54	809.4		[41]
GSs	PS	Polymerization	0.9 wt.%	41.42	2280		[125]
GO	PS	Solution process	2 wt.%	43.5	3580	1.3	[126]
FLG	PS	Solution process	0.9 wt.%	13.98		11.3	[127]
FLG	PS	Solution process	0.7 wt.%	16.03		17	[128]
fGNs	PS	Solution process	1 wt.%	78.2		2.38	[129]
GO	PS	Solution process	1.02 wt.%	13.60		1.185	[130]
MLG	PVC	Polymerization	0.3 wt.%		50.5		[131]
GNP	PVC	Solution process	2.5 wt.%	24	11.42	33.5	[132]
mGO	PVC	Solution process	5 wt.%	37.87	1694.23	2.61	[133]
rGO	PVC	Melt mixing	1 wt.%	16.16	46.9	362	[134]
rGO	PVC	Solution process	0.2 wt.%	35	1700	5	[135]
GO	PA6	Polymerization	0.1 wt.%	123	722	269	[71]
fGO	PA6	Solution process	1 wt.%		1900		[136]
fGr	PA6	Polymerization	0.2 wt.%	68.4		100	[137]
GO	PA6	Polymerization	0.65 wt.%	64.9			[138]
fGr	PA6	Polymerization	0.1 wt.%	500		20.7	[139]

## Data Availability

Data presented in this study are available on request from the corresponding author.

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
