# Peer review of "Graphene/Polymer Nanocomposites: Preparation, Mechanical Properties, and Application"

_polymers, 2022, doi:10.3390/polym14214733_

Round 1
Reviewer 1 Report
In-Yup Jeon et al. in their manuscript entitled “Graphene/Polymer Nanocomposites: Preparation, Mechanical Properties, and Application”, reviewed the preparation processes of the graphene/polymer nanocomposites, their mechanical properties, and their applications. However, some deficiencies listed below are observed:
1. The manuscript contains some typos/spelling mistakes. They should be corrected.
2. The introduction section is weak. The authors should add more information about the graphene and graphene/polymer composites in comparison with other carbon fillers, and their cons and pros.
3. It would be better if the authors added a table of all considered methods for obtaining graphene with their advantages and disadvantages.
4. The authors should rewrite the title of “3. Preparation” as the preparation of graphene/polymer composites.
5. Moreover, this section “3. Preparation” is weak. The authors should add more information about the preparation parameters, examples, cons and pros, with updated references.
6. It would be better if the authors can add a section of analysis techniques for graphene/polymer composites.
7. In the “4. Mechanical Properties” section, the authors wrote that graphene have been used to improve the mechanical performance of pure polymers by increasing the interaction between polymers and carbon materials. However, the addition of graphene can also lead to a decrease in the mechanical properties of graphene/polymer composites. The authors should mention this and add the appropriate references.
8. Based on reference [86], the authors presented that the addition of graphene led to improve the mechanical properties of the UHMWPE/(TrGO) composites. However, the improvement in the mechanical properties was not only associated with the addition of graphene, but also highly depended on the orientation process. The authors should mention this and discuss how the addition of filler can affect the draw ratio.
9. In Table 1, there are some cited references without a description in the main text, for example reference 113, 115. The authors should add the main properties of the composites in the main text of the manuscript.
10. The section of “6. Future perspectives” is weak. The authors should add their point of view in this section.
11. The conclusion should be rewritten. The authors should summarize the main points in their review in detail.
12. A discussion of how graphene affects the structure of the polymer (crystallization kinetics, crystallinity, crystals size, melting and crystallization temperature, etc.) in graphene/polymer composites is absent. The authors should add a section and discuss this important information. This information is required to explain the improvement in the mechanical properties of the graphene/polymer composites. Moreover, the authors should add a discussion about the used techniques for the investigation of the composites structure, such DSC, DMA, etc.
13. The DOI links for all references are absent; the authors should add them. Moreover, the references should be updated.
Reviewer 2 Report
This review of graphene/polymer nanocomposites is focusing on preparation and compatibilization methods, as well as possible applications.
A few suggestions for further improving:
Line 22: "their inadequate physical properties" --> Most readers would understand the point, though I would reformulate this, as whether adequate or not depends on the application and indeed certain polymer properties (e.g. insulation) make them actually perhaps the best cantidates.
Lines 10 and 24: "relieve restrictions" sounds inaccurate; probably something in the sense of "reaching the desired requirements" could be more clear?
Line 26: To be exact according to the definition, <100 nm in one dimension.
Line 71: Please split the sentence into two after "etc".
Section 3.2, Table 1, Section 4.2: There are more recent efforts about improving the dispersion of graphene in polypropylene in the melt, including publications in Polymers and Nanomaterials in the last three years, 2019-2022.
Round 2
Reviewer 1 Report
The revised manuscript is better. However, I still feel that the discussion of how graphene affects the structure of the polymer is weak.